# Development of Shape Prediction Model of Microlens Fabricated via Diffuser-Assisted Photolithography

**DOI:** 10.3390/mi14122171

**Published:** 2023-11-29

**Authors:** Ha-Min Kim, Yoo-Kyum Shin, Min-Ho Seo

**Affiliations:** 1School of Biomedical Convergence Engineering, Pusan National University, 49 Busandaehak-ro, Mulgeum-eup, Yangsan-si 50612, Republic of Korea; gkalsrla@pusan.ac.kr; 2Department of Information Convergence Engineering, Pusan National University, 49 Busandaehak-ro, Mulgeum-eup, Yangsan-si 50612, Republic of Korea; dbruadl44@pusan.ac.kr

**Keywords:** microlens array, diffuser-assisted photolithography, prediction model, photolithography

## Abstract

The fabrication of microlens arrays (MLAs) using diffuser-assisted photolithography (DPL) has garnered substantial recent interest owing to the exceptional capabilities of DPL in adjusting the size and shape, achieving high fill factors, enhancing productivity, and ensuring excellent reproducibility. The inherent unpredictability of light interactions within the diffuser poses challenges in accurately forecasting the final shape and dimensions of microlenses in the DPL process. Herein, we introduce a comprehensive theoretical model to forecast microlens shapes in response to varying exposure doses within a DPL framework. We establish a robust MLA fabrication method aligned with conventional DPL techniques to enable precise shape modulation. By calibrating the exposure doses meticulously, we generate diverse MLA configurations, each with a distinct shape and size. Subsequently, by utilizing the experimentally acquired data encompassing parameters such as height, radius of curvature, and angles, we develop highly precise theoretical prediction models, achieving R-squared values exceeding 95%. The subsequent validation of our model encompasses the accurate prediction of microlens shapes under specific exposure doses. The verification results exhibit average error rates of approximately 2.328%, 7.45%, and 3.16% for the height, radius of curvature, and contact angle models, respectively, all of which were well below the 10% threshold.

## 1. Introduction

Microlens arrays (MLAs) have attracted significant interest from diverse industries and academia owing to their wide range of applications, including optical and image sensors, display components, and microscopes [1,2,3,4,5,6,7,8,9]. The shape, dimensions, and array of the MLA are considered highly important parameters because they directly influence the optical path through the MLA, subsequently impacting the device performance [10,11,12]. Hence, it is crucial to fabricate the designed MLA with a specific shape and dimensions to ensure high accuracy and reliability for the intended application.

Various methods have been developed for MLA fabrication [13,14,15,16]. One notable approach is the “direct method”, which allows MLA production without intermediate steps. This method involves using a thermally stable dome-shaped template for MLA fabrication through hot embossing and inkjet printing to directly form an MLA on a substrate [15,17,18]. In addition, a thermal reflow method based on conventional photolithography has been employed [19,20,21]. These direct methods are advantageous in terms of cost-effectiveness and simplicity. However, concerns naturally arise regarding the low yield, fabrication reliability, and limited fill factor, especially in large-area MLA production. Therefore, “indirect methods” have been proposed as alternatives. In this method, a concave-shaped mold is fabricated using wet etching and soft lithography [2,22,23]. Subsequently, the MLA was fabricated using a molding-replication process on a mold. These approaches successfully demonstrated the high fabrication yield and reproducibility of MLA. However, these methods have limitations in terms of size and shape adjustability with high simplicity.

In this context, researchers developed a method called diffuser-assisted photolithography (DPL) that combines conventional lithography with optical diffuser sheets [24]. The DPL process utilizes a diffuser sheet to randomly scatter incident light, resulting in the patterning of lens-shaped 3D structures in a photoresist (PR). The density and shape of the MLA can be effectively controlled by designing a photomask. DPL exhibits superior adjustability of size and shape, a high fill factor, enhanced productivity, and excellent reproducibility, making it an appealing technique for MLA fabrication.

Despite these advantages, DPL-based MLA fabrication faces several challenges. Random changes in the optical path and intensity (or dose equivalent) of light passing through the diffuser sheet make it extremely difficult to predict the final shape and height of the microlens. In this regard, researchers have made considerable efforts to demonstrate specifically designed MLA through repetitive trial-and-error experiments, which consume significant time and cost [25]. Consequently, there is an urgent need to develop a method for predicting the final shape and dimensions of microlenses using the DPL method with high simplicity and accuracy.

Herein, we introduce a theoretical model designed to predict the microlens shape in response to varying exposure doses during the DPL process. We established a reliable MLA fabrication method capable of shape modulation in alignment with conventional DPL techniques. Through precise adjustments of the exposure dose, we generated an array of MLA configurations, each exhibiting a distinct shape and size. Subsequently, using experimentally acquired data encompassing the height, radius of curvature, and angles, we meticulously evaluated and developed precise theoretical prediction models to elucidate the relationship between the microlens shape and exposure dose. Subsequent validation of our model involved the accurate prediction of the microlens shape under specific exposure doses. Our validation process included experimental verification, which affirmed the exceptional accuracy of the model and highlighted its remarkable precision.

## 2. Materials and Methods

To obtain various lens shape-associated data, we first developed a microlens fabrication process (Figure 1). First, a 50 nm thick layer of chromium (Cr) was deposited on a glass wafer using an e-beam evaporator. Subsequently, a microscale photoresist (PR, AZ 9260, K1solution, Seoul, Republic of Korea) pattern was formed on the specimen using conventional photolithography. The positive photoresist (PR), AZ P4620 (K1solution, Republic of Korea), is spin-coated at 1200 rpm for approximately 60 s (with a 10 s acceleration), resulting in a coating of approximately 15 µm. Soft baking was performed at 70 °C for 30 min in an oven. Exposure was performed using a mask aligner with an exposure energy (*E*) of 400 mJ/cm^2^. After exposure, development was performed using MIF 300 (10454220521; Merck, Seoul, Republic of Korea) for 200 s, followed by rinsing with deionized (DI) water. Subsequently, Cr etching was performed to form an embedded mask. The Cr layer was patterned using wet chemical etching (Cr etchant; Sigma-Aldrich, Seoul, Republic of Korea). The Cr etchant (651826, Sigma-Aldrich, Republic of Korea) was diluted with DI water at a 1:5 ratio, and the etching process was conducted for 15 min at 24 °C. Subsequently, a Cr-embedded mask was formed on the glass wafer after stripping the PR pattern. Subsequently, a thick PR layer was applied to form the MLA. On top of this mask, a high-viscosity negative photoresist, su-8 3025 (K1solution, Republic of Korea), was spin-coated at 500 rpm for 5 s, followed by 1000 rpm for 30 s, resulting in a thickness of approximately 80 µm. Soft baking was performed at 95 °C for 30 min on a hot plate. We then proceeded with the backside lithography. Subsequently, a sandblasted diffuser was placed on the glass wafer, and exposure was conducted according to the exposure conditions. After exposure, PEB was performed on a hot plate at 95 °C for 3 min. Finally, the MLA was obtained by developing a specimen after post-exposure baking. The development was carried out in a su-8 developer (K1solution, Republic of Korea) for approximately 11 min, followed by rinsing with DI water and drying with N_2_ gas. It is worth noting that the shape of the MLA can be modulated by varying the *E* irradiated on the specimen in “Step 7” (Figure 1).

## 3. Results

To experimentally confirm the proposed fabrication, we first fabricated MLA using the proposed fabrication process. We first utilized an exposure energy, E = 40 mJ/cm^2^, for negative PR curing (step 7 in Figure 1), and the fabrication results were observed using an optical microscope (OM) and a scanning electron microscope (SEM). Regular embossed micropatterns were observed in the OM image (Figure 2A). Significant structural artifacts, such as defects and non-uniformity, were not observed over a large area, indicating the reliability of this process. We also fabricated other MLA shapes by modulating the exposure energy (E) in “step 7” from 30 to 260 mJ/cm^2^ and confirmed that different MLA shapes can be reliably fabricated via the proposed fabrication process. Figure 2B–E show the SEM images of the MLAs fabricated at E = 30, 40, 100, and 260 mJ/cm^2^, respectively.

Next, we obtained the dimensional data of different MLAs with respect to the exposure energy (E). We first defined the dimensional parameters, such as the height (h), radius of curvature of the lens (*k*), and contact angle (θ), determining the lens shape based on a conventional spherical cap model (Figure 3A) [15]. We then obtained the experimental h, k, and θ of the MLA fabricated by E = 30, 50, 100, 140, and 260 mJ/cm^2^. Figure 3B–D presents the measured h, *k*, and θ data from different lenses (n = 50), respectively. First, we confirmed that h of the MLA increases as *E* increases. Specifically, when *E* = 30 mJ/cm^2^, *h* is approximately ~30 μm, but *h* continuously increases and is saturated as E≥ 100 mJ/cm^2^. We attributed these results to the limited thickness of the PR. When E is higher than 100 mJ/cm^2^, the incident UV light penetrates through the end of the deposited PR; thus, it fully cures the deposited PR (thickness, *t* = 80 μm) and cannot make the lens shape of the exposed part of PR, finally creating an inverse trapezoidal shape. In the case of k, a decreasing trend is observed as E increases. Numerically, when E is 30 mJ/cm^2^, *k* has a value of approximately 22.5 nm^−1^. As E increased, k gradually decreased, and at approximately 140 mJ/cm^2^ or higher, the radius of curvature of the lens converged to zero, forming an inverse-trapezoidal shape. The measured θ of MLA also showed a similar trend by the increasing E. When the E is small, i.e., E = 30 mJ/cm^2^, the MLA has a small θ by ~30°, but it notably increases as E increases and reaches ~110° as E≥~200 mJ/cm^2^. It is worth noting that θ is also saturated when the E≥~200 mJ/cm^2^ because of the diffraction limitation in the embedded Cr mask. The measured h, k, and θ results show their own characteristics in the shape and dimension with respect to E; however, they are highly reproducible, and trends of the measured data are well explained using the conventional mechanism of DPL; thus, the data seem to be appropriate for application in the development of the shape prediction model based on the DPL.

## 4. Discussions

Based on the measured dimensional data, we made efforts to determine numerical models that can predict the dimensional parameters, such as h, k, and θ. To develop the prediction model, we consider representative statistical models, such as “Exponential Regression, Lognormal regression, and Boltzmann regression”. (i) “Exponential regression” is a statistical modeling technique used to analyze data that exhibit an exponential growth or decay pattern. In exponential regression, the relationship between the independent variable (often denoted as “*x*”) and the dependent variable (“*y*”) is modeled using the following exponential function:$$y=y_{0}+Ae^{\pm E/t}$$
where $y_{0}$, *A*, and *t* are the experimental fitting parameters. Exponential regression is commonly used in various fields to model data with exponential relationships such as population growth, bacterial growth, financial investments, and radioactive decay. The goal of exponential regression is to estimate the values of “a” and “b” that best fit the given data, allowing for predictions and an understanding of the underlying growth or decay process. (ii) Lognormal cumulative distribution function (CDF) regression is a probabilistic regression analysis technique, particularly when a continuous variable follows a lognormal distribution. This model is representatively utilized for a probability distribution that follows a normal distribution after taking its natural logarithm. Because the log-transformed values follow a normal distribution, the lognormal distribution is commonly used to model continuous variables that are strictly positive as follows:$$y=y_{0}+A\int_{0}^{E}\frac{1}{\sqrt{2\pi }\omega x}\;e^{-\frac{{\left(\ln\left(x\right)-x_{c}\right)}^{2}}{2\omega^{2}}}dx$$
where, $y_{0}$, *A*, *x_c_*, and ω are the experimentally fitting parameters. Lognormal regression is commonly used in various fields, including finance, environmental science, biology, and healthcare, where data often exhibit skewed distributions with positive values. This allows the model and data analysis with a lognormal pattern to be more accurate than linear regression, which assumes a normal distribution of the dependent variable. (iii) “Boltzmann regression”, also known as Boltzmann sigmoidal regression, is a modeling technique used to fit data to a sigmoidal or S-shaped curve. This equation is defined as follows:$$y=A_{2}+\frac{\left(A_{1}-A_{2}\right)}{1+e^{\frac{\left(E-E_{0}\right)}{dx}}}$$
where $A_{1}$, $A_{2}$, $E_{0}$, and dx are the fitting parameters. This regression method is commonly applied in various scientific fields, including physics, chemistry, biology, and neuroscience, to describe the relationships between variables that exhibit sigmoidal behavior. Boltzmann regression is a valuable tool for understanding and quantifying sigmoidal relationships in data. This allows researchers to model and predict the behavior changes of a system changes as a function of an independent variable, which is essential for various scientific and engineering applications. Subsequently, in accordance with the representative statistical model, we optimized the fitting parameters to obtain a dimensional parameter prediction model that aligned closely with the experimental data.

Figure 4 shows the height results for each model fitted to the exposure dose. Each graph was arranged in ascending order of how well the model explained the variables. For the Boltzmann model, when the fitting parameters are $A_{1}=-23,770\;\mu m$, $A_{2}=84.99\;\mu$m, $E_{0}=-125.41$ mJ/cm^2^, and dx=25.876 mJ/cm^2^ (see the equation above), it shows an explanatory power of 96.8%. Similarly, the Lognormal CDF model exhibits an explanatory power of 95.4% when the parameters are $y_{0}$ = −18.713 μm, $A=103.94\;\mu m,\;x_{c}=3.508,\;and\;\omega =0.633$. Finally, for the exponential decay model, when the parameters are $y_{0}$ = 84.994 μm, *A* = −186.628 μm, and *t* = 25.919 mJ/cm^2^, it demonstrates an explanatory power of 97.8%. Most models showed an excellent explanatory power for the height, with the highest result observed in the exponential decay model, indicating that it was the most accurate predictive model among the proposed models.

Figure 5 shows the radius of curvature for each model fitted to the exposure dose. Similarly, each graph was arranged in ascending order of how well the model explained the variables. In the case of the Lognormal CDF model, the fitting parameters $y_{0}$ = 0 μm^−1^, *A* = 0.02 μm, *x*_c_ = 4.34, and ω = 0 resulted in a negative R-squared value, indicating an inability to explain the data. For the exponential decay model, when the parameters are $y_{0}$ = −2822.30 μm, *A* = 2822.32 μm, and *t* = $2.8\times 10^{7}$ mJ/cm^2^, it exhibits an explanatory power of 75.1%. Finally, for the Boltzmann model, when the parameters are $A_{1}$ = 0.02158 μm, $A_{2}=-3.8867\times 10^{-4}\;\mu m$, $E_{0}=93.63$ mJ/cm^2^, and dx = 14.53 mJ/cm^2^, it exhibits an explanatory power of 99.6%. Among the models for predicting the radius of curvature, the Lognormal CDF model failed to make predictions, whereas the Boltzmann model showed the highest explanatory power, making it the most accurate predictive model among the proposed models.

Figure 6 shows the contact angle results for each model fitted to the exposure dose. Each graph was arranged in ascending order of how well the model explained the variables. For the Boltzmann model, when the fitting parameters are $A_{1}$ = −201,289°, $A_{2}=$102.61°, $E_{0}=-$154.35 mJ/cm^2^, and dx = 23 mJ/cm^2^ (see the equation above), it exhibits an explanatory power of 96%. Similarly, the exponential decay model exhibits an explanatory power of 96% when the parameters are $y_{0}$ = −102.61°, *A* = −245.72°, and *t* = 23.02 mJ/cm^2^. Finally, for the Lognormal CDF model, when the parameters are $y_{0}$ = −6,126,473°, *A* = 6,126,583°, *x*_c_ = −10.19, and ω = 3.219, it exhibits an explanatory power of 98.1%. Most models showed excellent explanatory power for contact angle, with the highest result observed in the Lognormal CDF model, indicating that it was the most accurate predictive model among the proposed models.

Finally, the accuracy of the developed models was estimated using untrained data (validation). For the validation, we further fabricated four more MLAs using E = 40 mJ/cm^2^, 80 mJ/cm^2^, 150 mJ/cm^2^, and 200 mJ/cm^2^ and plotted the measured h, *k*, and θ of the MLA on the developed models (Figure 7A,D,G,J). First, in the case of *h*, the average height of the 40 mJ/cm^2^ sample was 43.69 μm, and the predicted height through the model was 43.526 μm. The average error rate was approximately 0.375%. For the 80 mJ/cm^2^ sample, the average height was 72.01 μm, whereas the calculated value was 73.93 μm, resulting in an average error rate of 2.67%. For the 150 mJ/cm^2^ sample, the average height was 81.87 μm, whereas the calculated value was 84.08 μm, resulting in an average error rate of 2.7%. For the 200 mJ/cm^2^ sample, the average height was 82.267 μm, and the calculated value was 85.19 μm, resulting in an average error rate of 3.55%. Therefore, the average error rate observed in the test set was approximately 2.328%, indicating a low deviation. Secondly, for the curvature radius, *k*, in the case of the 40 mJ/cm^2^ sample, the average curvature radius was 21.2 nm^−1^, and the calculated value was 21.1 nm^−1^, showing a very small difference. The average error rate at this lithography intensity was 0.47%, which is extremely low. For the 80 mJ/cm^2^ sample, the average curvature was 14.9 nm^−1^, and the calculated value was 12.75 nm^−1^, resulting in an average error rate of 14%. The error rate was greater than 10%; however, it was a very small unit, and it was a very small difference. For the 150 mJ/cm^2^ and 200 mJ/cm^2^ samples, the curvature radius takes on an infinite value due to the trapezoidal shape; thus, it is close to 0 μm^−1^. The calculated results also appear to be close to 0. The error rate is approximately 0. Therefore, the average error rate of the curvature radius model was approximately 7.45%. Finally, the contact angle model was validated. For the 40 mJ/cm^2^ sample, the average contact angle was 66.5°, whereas the angle calculated using the model was 60.2336°. The error rate was 9.4%. For the 80 mJ/cm^2^ sample, the average contact angle was 90.59°, whereas the calculated value was 991.664°, resulting in an average error rate of 1.18%. Third, when exposed to a power of 150 mJ/cm^2^, the average contact angle was 104.21°, and the predicted value was 103.85°, which has an error rate of 0.35%. In the case of the test sample with a lithography intensity of 200 mJ/cm^2^, the average contact angle was 105.1°, and the calculated value using the model was 106.87°. In this case, the error rate was approximately 1.68%. The average error rate of the contact angle model was approximately 3.16%. In conclusion, the validation results for the three variable models showed average error rates of less than 10%, indicating good predictive accuracy.

## 5. Summary and Conclusions

In summary, we focused on developing predictive models of microlenses fabricated via diffuser-assisted photolithography (DPL). DPL is an innovative method that combines conventional lithography with an optical diffuser sheet to create 3D lens structures within photosensitive materials. The appeal of DPL lies in its versatility, which enables control over the size and shape, high fill factor, productivity, and reproducibility in microlens array (MLA) production. Despite these advantages, the inherent randomness of light scattering through the diffuser sheet makes it difficult to accurately predict the final shape and dimensions of the microlenses. This has led researchers to rely heavily on trial-and-error experiments, which are resource-intensive approaches.

Our research sought to bridge this gap by developing precise and user-friendly prediction models for the microlens dimensions and shapes within the DPL process. The proposed methodology was divided into several steps. (I) Fabrication Method: Our journey began with the evaluation of a reliable fabrication method capable of consistently producing microlenses with specific dimensions and shapes in response to varying exposure energy levels. This initial investigation laid the foundation for the subsequent data collection and model development. (II) Data Collection and Model Development: To establish our prediction models, we undertook extensive data collection, generating a dataset of 250 microlenses. The main dimensional variables that will be variables in subsequent predictive model development are height (h), radius of curvature (k), and angle of contact (θ). (III) Model Development and Selection: With our dataset, we embarked on the development of prediction models for each of the identified dimensional parameters. Our approach involved testing several statistical models, each tailored to its respective parameters. After rigorous optimization of the model parameters, we selected the exponential decay model for the height, the Boltzmann model for the radius of curvature, and the Lognormal CDF model for the contact angle. These models exhibited remarkable accuracy, with explanatory powers of 97.8% for the height, 99.6% for the radius of curvature, and 98.1% for the contact angle. (IV) Validation and Model Performance Assessment: To assess the practical effectiveness of our predictive models, we conducted a validation phase. Four additional MLAs were fabricated using distinct exposure energy levels of 40, 80, 150, and 200 mJ/cm^2^. Subsequently, we compared the actual measurements of the height, radius of curvature, and contact angle from these MLAs with the predictions generated using our models. For the height, the combined mean error rate of the four datasets was approximately 2.328%, indicating a high point density in the model prediction. In terms of the radius of curvature, the average error rate of the curvature radius model was approximately 7.45%. Finally, the average error rate for the contact angle model was approximately 3.16%. To further validate the predictive power of our models, we visually compared the lens shapes projected by our models using the validation dataset with actual lens shapes observed through scanning electron microscope (SEM) images. This visual assessment confirmed that our models closely approximated real-world lenses, thereby reinforcing their accuracy and reliability.

In conclusion, our study contributed significantly to the field of microlens production and modeling. Our research approach effectively eliminated the need for extensive trial-and-error experiments and saved valuable time and resources to maximize efficiency. It also facilitated the creation of custom microlenses for a wide range of applications. The implications of our study extend across several industries, including optics, image sensors, microscopes, and displays, where the precise control of MLA characteristics directly affects device performance. By further refining and extending the applicability of future predictive models, we can go beyond what can be achieved using current optical technologies. In addition, collaboration between academia and industry can accelerate the advancement of fields relying on microlens technologies.

## Figures and Tables

**Figure 1 micromachines-14-02171-f001:**
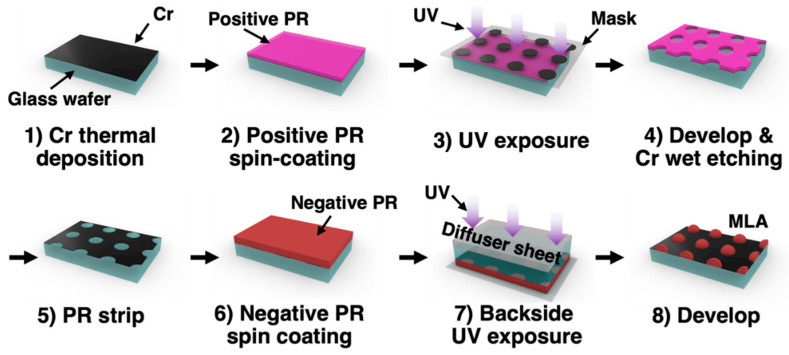
Fabrication process of proposed MLA based on the DPL.

**Figure 2 micromachines-14-02171-f002:**
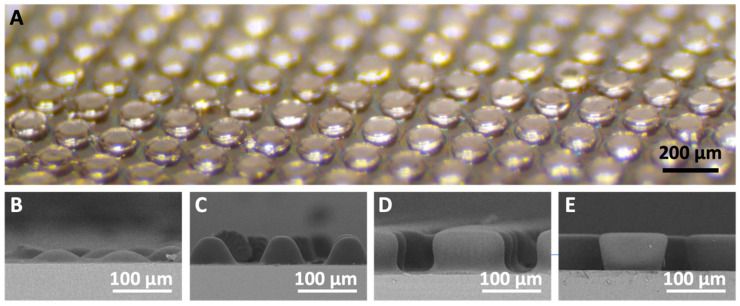
Fabrication results of various MLAs. (**A**) Optical microscopy image of MLA. (**B**–**E**) SEM images of MLA according to exposure dose: (**B**) 30 mJ/cm^2^, (**C**) 40 mJ/cm^2^, (**D**) 100 mJ/cm^2^, and (**E**) 260 mJ/cm^2^.

**Figure 3 micromachines-14-02171-f003:**
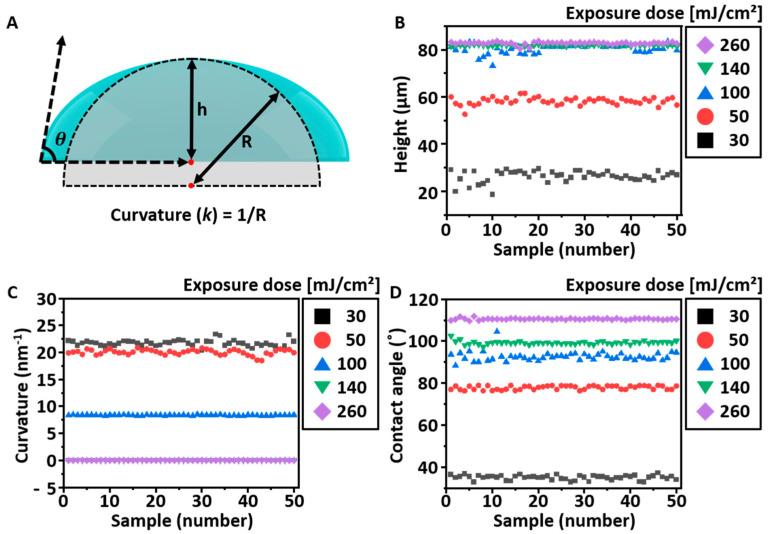
Measured angle and height data of the fabricated MLAs. (**A**) Schematic illustration of parameters for the 3D MLA model (h: height of MLA, R: radius of curvature of MLA, and θ: contact angle of MLA). Distribution of (**B**) heights, (**C**) curvatures, and (**D**) contact angles for measured samples of the fabricated MLAs according to exposure doses (mJ/cm^2^).

**Figure 4 micromachines-14-02171-f004:**
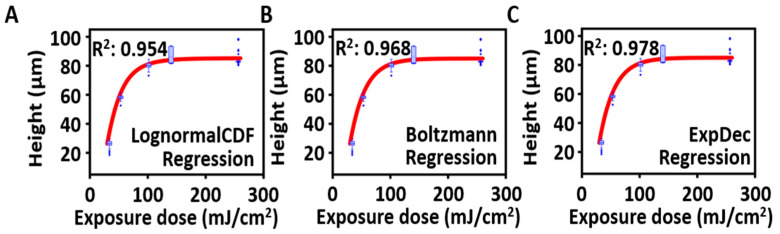
Derivation and verification of the proposed height prediction model. To identify the most suitable model for describing the height, representative regression attempts were conducted as follows: (**A**) Boltzmann regression performed (explained variance: 95.4%), (**B**) Lognormal CDF regression (explained variance: 96.8%), and (**C**) one-phase exponential decay regression (explained variance: 97.8%). (Blue box plot: the height distribution for each exposure, Red line: the regression model fitted).

**Figure 5 micromachines-14-02171-f005:**
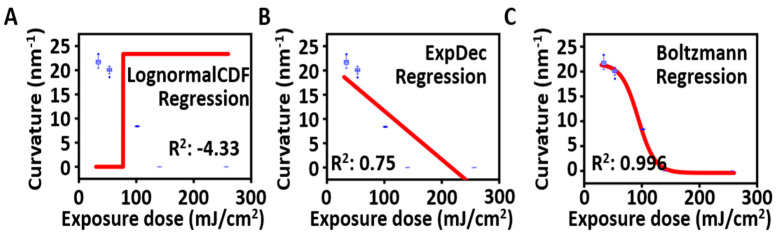
Derivation and verification of the proposed curvature prediction model. Various regression analyses were conducted to find a model explaining the radius of curvature with respect to the exposure dose: (**A**) Lognormal CDF regression (no explained variance). (**B**) One-phase exponential decay regression (explained variance: 75%). (**C**) Boltzmann regression (explained variance: 99.6%). (Blue box plot: the curvature distribution for each exposure, Red line: the regression model fitted).

**Figure 6 micromachines-14-02171-f006:**
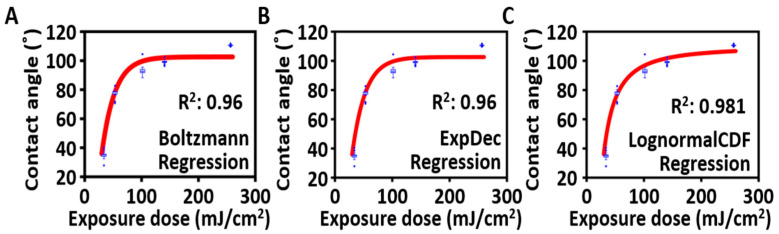
Derivation and verification of the proposed contact angle prediction model. Various regression analyses were conducted to find a model explaining the change in contact angle with respect to the exposure dose: (**A**) Boltzmann regression (explained variance: 96%). (**B**) One-phase exponential decay regression (explained variance: 96%). (**C**) Lognormal CDF regression (explained variance: 98.1%). (Blue box plot: the contact angle distribution for each exposure, Red line: the regression model fitted).

**Figure 7 micromachines-14-02171-f007:**
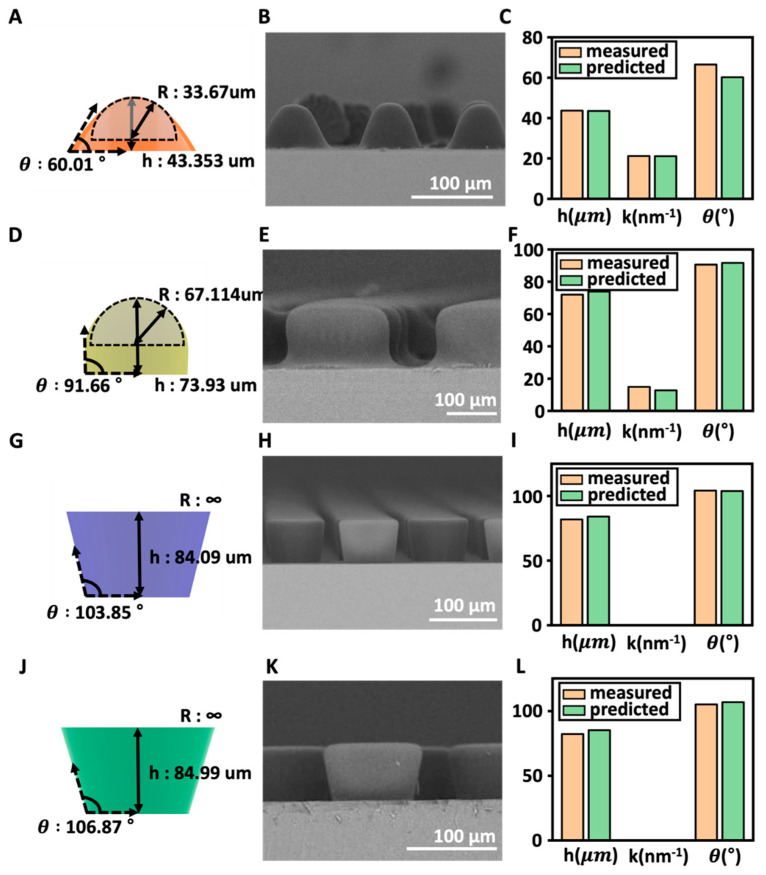
The performance of the model validated with the validation dataset and simulation. (**A**) Predictive models for each variable at E = 40 mJ/cm^2^, (**B**) SEM image of the actual sample at E = 40 mJ/cm^2^ and (**C**) at E = 40 mJ/cm^2^, displaying actual values and predicted values for height, radius of curvature, and contact angle (orange: actual measurements; green: predicted values). (**D**) Predictive models for each variable at E = 80 mJ/cm^2^, (**E**) SEM image of the actual sample at E = 80 mJ/cm^2^ and (**F**) at E = 80 mJ/cm^2^, displaying actual values and predicted values for height, radius of curvature, and contact angle (orange: actual measurements; green: predicted values). (**G**) Predictive models for each variable at E = 150 mJ/cm^2^, (**H**) SEM image of the actual sample at E = 150 mJ/cm^2^ and (**I**) at E = 150 mJ/cm^2^, displaying actual values and predicted values for height, radius of curvature, and contact angle (orange: actual measurements; green: predicted values). (**J**) Predictive models for each variable at E = 200 mJ/cm^2^, (**K**) SEM image of the actual sample at E = 200 mJ/cm^2^ and (**L**) at E = 200 mJ/cm^2^, displaying actual values and predicted values for height, radius of curvature, and contact angle (orange: actual measurements; green: predicted values).

## Data Availability

The data that support the findings of this study are available from the corresponding author upon reasonable request.

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
