# Peer review of "Development of Shape Prediction Model of Microlens Fabricated via Diffuser-Assisted Photolithography"

_micromachines, 2023, doi:10.3390/mi14122171_

Round 1

Reviewer 1 Report

Comments and Suggestions for Authors

1.This paper has developed a manufacturing method for diffuser-assisted photo-lithography, by introducing a theoretical model, the density and shape of the micro-lens array can be effectively predicted. A large amount of space in the article is used to extract the parameters of microlens and model building, while there is less verification of model accuracy. The author can consider increasing other exposure energy mentioned above for verification.

2.The author said that "when the E=30 mJ/cm2, the h is about ~20 μm" for figure 3B, but I look the h is closer ~30 μm, please confirm it. 

3.Conclusions and summaries are complicated and not concise enough.

4.The format of the article needs to be carefully checked. The annotation of the picture is not consistent with the case of the picture title. Formula should be centered in the text and the format of references is not uniform.

Comments on the Quality of English Language

Lanuguage is easy to understand but it is suggested to be improved more suitable for scientific reading. 

Author Response

The authors would like to express our sincere appreciation for your editorial efforts and your kind response to our manuscript. We have received the letter containing valuable comments from the reviewers, and their insightful reviews have been truly impressive and beneficial to us. We recognize that this editorial process contributes to improving the quality of our paper.

We have diligently addressed all of the reviewers' questions and have made corresponding revisions to our manuscript, which is attached as word file. Additionally, we have enlisted the services of Editage to enhance the manuscript's conciseness and comprehensiveness for our readers. All amended sentences were underlined here and marked in the revised manuscript.

Reviewer 2 Report

Comments and Suggestions for Authors

I thoroughly review the paper entitled “Development of Shape Prediction Model of Micro-lens Fabricated by Diffuser-assisted Photolithography ". The authors introduced a theoretical model capable of predicting the outcomes of diffuser-assisted photolithography, and its accuracy was also experimentally validated. Therefore, I recommend this paper for publication in Micromachines, with the suggestion of minor revisions to further enhance its clarity.

1. Additional explanation is needed regarding the reasons to use PI film in step 7 of the fabrication process. 

2. In Section 3 (Results), the authors need to check the expression 'step 1 in Figure 1' mentioned in line 3. Additionally, the authors should check whether the sentence in line 5—'Over a large area, significant structural artifacts and defects are observed, indicating the reliability of the developed process'—accurately represents the intended message.“

3. In Figure 7 c,f, does “a” represent the contact angle? The authors used theta throughout the paper, so I recommend using the theta than “a”.

Comments on the Quality of English Language

I recommend a comprehensive review and improvement of the English expression throughout your paper.

Author Response

(The authors gave the same response as above.)

Round 2

Reviewer 1 Report

Comments and Suggestions for Authors

The author has carefully revised it according to the comments and agreed to accept it.

Comments on the Quality of English Language

The author has refined and streamlined the English writing based on revisions.